# An Electrochemically Prepared Mixed Phase of Cobalt Hydroxide/Oxyhydroxide as a Cathode for Aqueous Zinc Ion Batteries

Fuwei Li [1,2], Yunbo Zhu [1,2], Hiroshi Ueno [3,4] and Ting Deng [1,2,*]

1   Key Laboratory of Automobile Materials of MOE, School of Materials Science and Engineering, Jilin University, Changchun 130012, China
2   Jilin Provincial International Cooperation Key Laboratory of High-Efficiency Clean Energy Materials, Jilin University, Changchun 130012, China
3   Creative Interdisciplinary Research Division, The Frontier Research Institute for Interdisciplinary Sciences (FRIS), Tohoku University, Sendai 980-8578, Japan; hiroshi.ueno.d5@tohoku.ac.jp
4   Department of Chemistry, Graduate School of Science, Tohoku University, Sendai 980-8578, Japan
*   Correspondence: tdeng@jlu.edu.cn

**Abstract:** Cobalt hydroxide is a widely studied electrode material for supercapacitor and alkaline zinc ion batteries. The large interlayer spacing of $Co(OH)_2$ is also attractive to store Zn ions. However, $Co(OH)_2$ is quite unstable in the acidic $ZnSO_4$ electrolyte due to its amphoteric nature. Herein, we synthesized a mixed phase of $Co(OH)_2/CoOOH$ via a two-step electrochemical preparation. As the cathode material for an aqueous zinc ion battery (AZIB), $Co(OH)_2/CoOOH$ delivered a maximum capacity of 164 mAh $g^{-1}$ at 0.05 A $g^{-1}$ and a high energy density of 275 Wh $kg^{-1}$. Benefiting from the low charge-transfer resistance, a capacity of 87 mAh $g^{-1}$ was maintained at 1.6 A $g^{-1}$, showing a good rate performance of the mixed phase. Various spectroscopy analyses and simulations based on the density functional theory (DFT) suggested a higher thermal stability of the mixed phase than pure $Co(OH)_2$, due to its less local structural disorder. The reduced Co-Co and Co-O shells increased the mechanical strength of the mixed phase to accommodate $Zn^{2+}$ ions and endure the electrostatic repulsion, resulting in an enhanced cycling stability. The mixed phased also delivered a good stability at the current density of 0.05 A $g^{-1}$. After 200 cycles, a capacity retention of 78% was retained, with high Coulombic efficiencies. These results provide a new route to synthesize high-performance LDH for aqueous zinc ion batteries.

**Keywords:** aqueous zinc battery; cobalt hydroxide; CoOOH; mixed phase

## 1. Introduction

The utilization of renewable energy sources, such as wind, solar, and tidal energies, is expected to alleviate the environmental pollution resulted from the combustion of fossil fuels. Due to the seasonal and regional factors, the storage of these energies with high efficiency, therefore, is of great significance to the wide application of these renewable energy sources. Among various energy storage systems, lithium ion batteries, due to their high energy density, are dominant in the markets of portable electronic devices and electric automobiles [1,2]. The safety issue and high cost of lithium ion batteries have driven us to search for alternative energy storage devices (ESDs) with comparable performance, but with a higher safety and lower cost. Aqueous ESDs, such as supercapacitor and nickel/metal hydride batteries, have been extensively investigated due to their excellent power densities and long lifespans. Unfortunately, the further application of these aqueous ESDs is restricted by the low energy density. Recently, aqueous zinc ion batteries (AZIBs) have drawn an exceptional amount of attention due to their environmental friendliness, high theoretical capacity of the Zn anode (820 mAh $g^{-1}$), and their abundancy in the Earth's crust [3,4]. However, the application of AZIBs is restricted by the unsatisfying

electrochemical performance of cathode materials. Currently, Mn- and V-based oxides are the most investigated cathode materials for AZIBs [5,6]. In addition to the low redox potential of V-based oxides, along with the dissolution problem of Mn-based oxides, their performance is also affected by the strong interaction between zinc ions and the crystal structure of oxides [7]. Therefore, seeking new cathode materials is still of great significance to develop AZIBs with high energy densities and long lifespans.

Layered double hydroxides (LDHs), with tunable chemical compositions and 2D pore structures, exhibit excellent pseudocapacitive activities in battery/supercapacitor hybrid ESDs [8–10]. Facilitated via the $H^+$ intercalation/deintercalation mechanism and the structural similarity, cobalt hydroxide ($Co(OH)_2$) has demonstrated an extraordinary performance in supercapacitors [11]. Its atoms are very exposed on the surface of its unique 2D morphology, and the large interlayer spacings facilitates the charge transfer between the electrolyte and the active material. However, the interlayer spacings of $Co(OH)_2$ are usually occupied by the anion residues ($NO_3^-$, $CO_3^{2-}$, organic anions, etc.) from the preparation to neutralize the positive charge of the Co-O layers [12]. As a result, Zn ion storage is prohibited by either the interlayer species or the electrostatic repulsion from the positively charged Co-O layers. Our previous study has proven that the excellent pseudocapacitive behavior originates from the rapid transformation between $Co(OH)_2$ and CoOOH based on their structural similarity [11]. Compared to $Co(OH)_2$, CoOOH not only preserves the morphology of $Co(OH)_2$, but also shows a higher level of thermal stability than $Co(OH)_2$, since each H atom is shared by two adjacent Co-O layers, and the positive charges are subsequently reduced, which presents CoOOH as a promising cathode material for AZIBs. Nevertheless, it remains uncertain to fabricate a high-performance CoOOH cathode for AZIBs, since no relevant study has been reported.

Herein, a mixed phase of $Co(OH)_2$/CoOOH was synthesized via a two-step electrochemical preparation. As an AZIB cathode material, $Co(OH)_2$/CoOOH showed a high average potential of 1.7 V vs. $Zn^{2+}$/Zn, which is higher than most of the reported cathode materials for AZIBs. Correspondingly, the mixed phase of $Co(OH)_2$/CoOOH can deliver a maximum capacity of 164 mAh $g^{-1}$ at 0.05 A $g^{-1}$ in 1 M $ZnSO_4$ electrolyte. The $Co(OH)_2$/CoOOH cathode also exhibited an excellent rate performance; a high capacity of 87 mAh $g^{-1}$ was obtained at 1.6 A $g^{-1}$. X-ray adsorption spectroscopy (XAS), along with theoretical simulations, indicated the high crystallinity and thermal stability of the mixed phase. The even distribution of CoOOH in the mixed phase suppresses the structure's degradation. The reduced Co-Co and Co-O shells increased the mechanical strength of the mixed phase to accommodate $Zn^{2+}$ ions and endure the electrostatic repulsion, resulting in an enhanced cycling stability. The mixed phase of $Co(OH)_2$/CoOOH showed a stable cycle life of 200 cycles at 0.05 A $g^{-1}$, with a capacity retention of 78%. The excellent performance of $Co(OH)_2$/CoOOH has demonstrated the promising future of LDHs as cathode materials for AZIBs.

## 2. Results

The mixed phase of $Co(OH)_2$/CoOOH was synthesized via a two-step electrochemical preparation, as illustrated in Figure 1A. Firstly, $Co(OH)_2$ was electrodeposited on the carbon fiber paper (CFP) as the substrate in a three-electrode system, in which a CFP, a platinum plate, a saturated calomel electrode, and a 1 M $Co(NO_3)_2$ solution served as the working electrode, counter electrode, reference electrode, and electrolyte, respectively. As the supercapacitor electrode, $Co(OH)_2$ loses one H atom and is then oxidized to CoOOH to store charges in alkaline media. Accordingly, the as-prepared $Co(OH)_2$ electrode was placed into a 1 M KOH solution for a cyclic voltammetry (CV) treatment to prepare the mixed phase of $Co(OH)_2$/CoOOH. The CV potential was set in the range of −0.1~0.45 V. To acquire the structural information, X-ray diffraction (XRD) was conducted on each step of the preparation. Figure 1B depicts the XRD patterns of the CFP after the electrodeposition, alkaline media contact, and the CV treatment, in which α, β, and C stand for α-$Co(OH)_2$, β-$Co(OH)_2$, and CoOOH, respectively. Two peaks were observed at 33° and 59°, which can

be indexed to the (100) and (110) planes of $\alpha$-Co(OH)$_2$ (PDF card No.460605). Compared to the pure CFP (Figure S1), the feature peak of the carbon materials (002) at 26° was far more reduced, indicating the well coating of $\alpha$-Co(OH)$_2$ on the CFP. The $\alpha$-Co(OH)$_2$ is known for its large interlayer spacing due to either anion or molecule insertion from the electrolyte. Our previous work has manifested that the electrochemically deposited $\alpha$-Co(OH)$_2$ was unstable, which could easily transform into $\beta$-Co(OH)$_2$ in alkaline media by removing the NO$_3^-$ species from the structure. The same phenomenon was also observed in this work. After being soaked in a 1 M KOH solution, the (001), (100), (101), (102), (110), and (111) planes of $\beta$-Co(OH)$_2$, with a high level of crystallinity, were detected in the red line (PDF card No.300443), while no signs of $\alpha$-Co(OH)$_2$ were detected in the XRD patterns, indicating the transformation from $\alpha$-Co(OH)$_2$ into $\beta$-Co(OH)$_2$. After the CV treatment, three new peaks were detected at 20°, 38°, and 65°, which can be indexed to the (001), (012), and (018) planes of CoOOH (PDF card No.070159), respectively, indicating the co-existence of $\beta$-Co(OH)$_2$ and CoOOH.

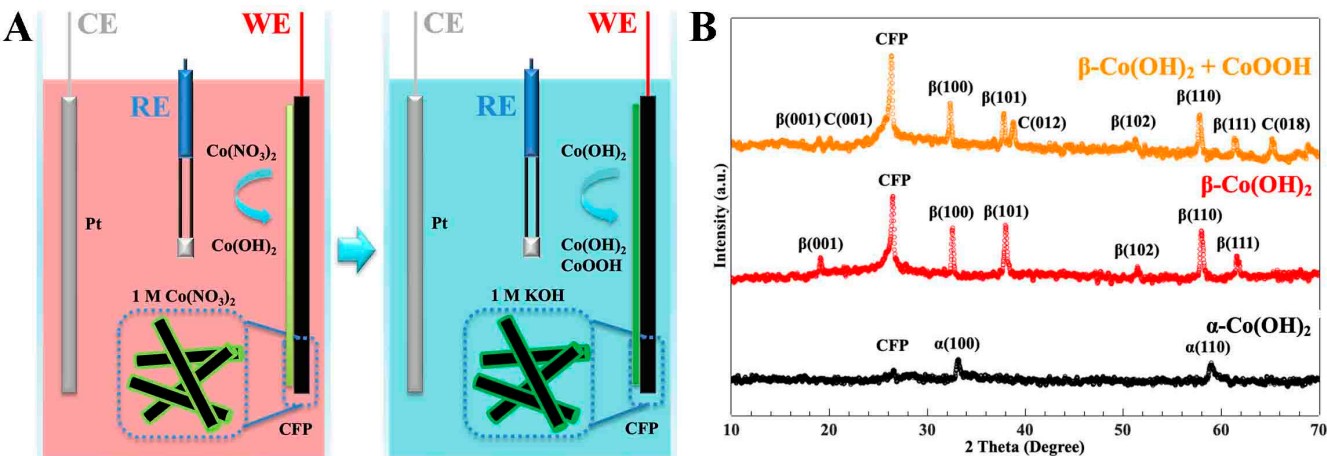

**Figure 1.** (**A**) The illustration of the synthesis of the Co(OH)$_2$/CoOOH mixed phase on the CFP. (**B**) The XRD patterns of $\alpha$-Co(OH)$_2$, $\beta$-Co(OH)$_2$, and the Co(OH)$_2$/CoOOH mixed phase on the CFP. Black line: $\alpha$-Co(OH)$_2$; red line: $\beta$-Co(OH)$_2$; orange line: Co(OH)$_2$/CoOOH mixed phase.

The morphology evolution of Co(OH)$_2$ was analyzed via scanning electron spectroscopy (SEM) and transmission electron spectroscopy (TEM). As shown in Figure 2A, the as-prepared $\alpha$-Co(OH)$_2$ displayed a petal-like morphology, with an average thickness of 200 nm. With a closer scrutiny, each petal was loosely stacked, consisting of multiple Co(OH)$_2$ layers with large interlayer spacings (Figure 2B). After being soaked in a 1 M KOH solution, the morphology of Co(OH)$_2$ changed dramatically from the loosely stacked petals to the hexagonal platelets, corresponding to the change in the XRD pattern (Figure 2C). In the hexagonal platelets, no obvious void was observed, suggesting a decreased interlayer spacing. In addition, TEM images also confirmed the morphology evolution from $\alpha$- to $\beta$-Co(OH)$_2$. The porous structure was observed in $\alpha$-Co(OH)$_2$ (Figure 2D), which disappeared and changed to the compact stack of Co(OH)$_2$ layers (Figure 2E), when the electrode was put into alkaline media. No obvious pores were identified. These results indicate that the actual reactant was $\beta$-Co(OH)$_2$. In addition, the space group of hexagonal $\beta$-Co(OH)$_2$ was R3m. The (010) and (100) crystal planes exhibited an acute angle of 60° (Figure 2F), also suggesting the formation of $\beta$-Co(OH)$_2$. After the CV treatment, the morphology of the mixed phase of $\beta$-Co(OH)$_2$ and CoOOH was quite similar to $\beta$-Co(OH)$_2$ (Figures S2 and S3), showing a morphology of well-shaped hexagonal platelets, which resulted from the similar structural parameters of $\beta$-Co(OH)$_2$ and CoOOH.

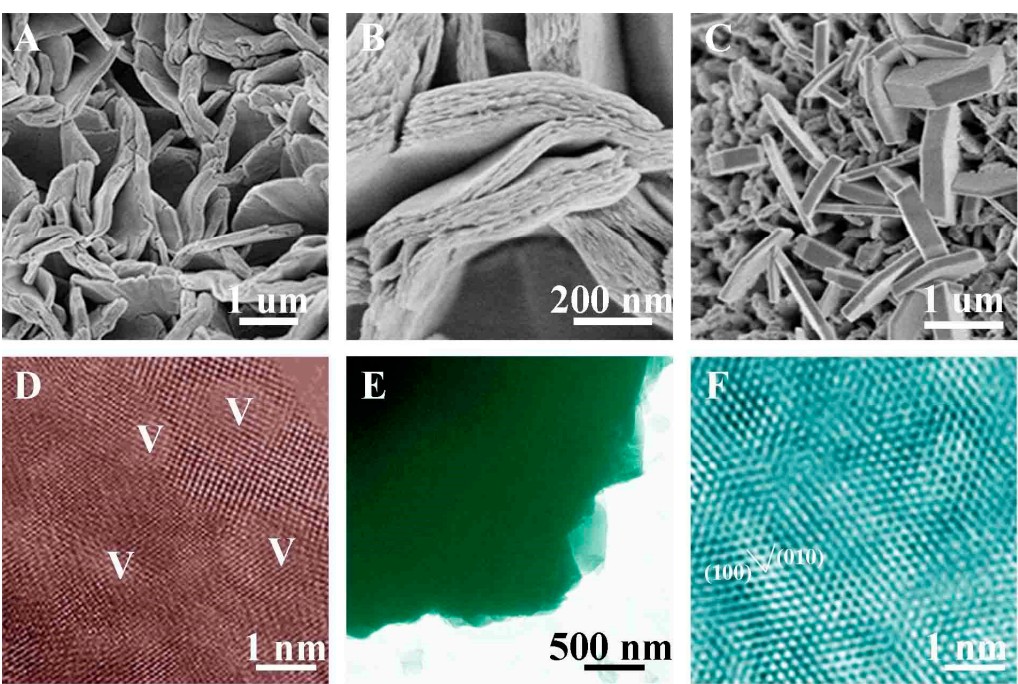

**Figure 2.** (**A**,**B**) The SEM images of α-Co(OH)$_2$ with different magnitudes. (**C**) The SEM image of β-Co(OH)$_2$. (**D**) TEM image of α-Co(OH)$_2$. (**E**,**F**) TEM and HRTEM images of β-Co(OH)$_2$.

Fourier transform infrared spectroscopy (FTIR) and X-ray photoemission spectroscopy (XPS) analyses were conducted to probe the surface state of the Co(OH)$_2$/CoOOH mixed phase. Figure 3A shows the FTIR spectra of the as-prepared Co(OH)$_2$ and the Co(OH)$_2$ after their immersion in the alkaline media. There was a sharp absorption band observed at 1383 cm$^{-1}$ in the spectrum of α-Co(OH)$_2$, representing the N-O vibration from the inserted NO$_3^-$ species [13]. The intensity of the N-O vibration was substantially reduced after diving the as-prepared Co(OH)$_2$ in the KOH solution, indicating the removal of NO$_3^-$. In addition, an intense band appeared at 3572 cm$^{-1}$ in the spectrum of β-Co(OH)$_2$, which is the typical feature of OH groups. The transformation from α- to β-Co(OH)$_2$ could be explained by the NO$_3^-$ replacement of OH$^-$ with a smaller size, leading to the compact arrangement of the Co-O layers. According to the FTIR analysis, α-Co(OH)$_2$ is quite unstable in the alkaline media, which can rapidly transform into β-Co(OH)$_2$. So, the actual reactant is β-Co(OH)$_2$, rather than α-Co(OH)$_2$, to store Zn$^{2+}$ ions. Figure 3B depicts the XPS spectrum of C 1s of α-Co(OH)$_2$ on the CFP. In addition to the typical signals of the sp$^2$ carbon and some surface function groups, such as C-O, C=O, and COOH, strong Co-O-C interactions were generated between the CFP and Co(OH)$_2$ at 294 eV and 298 eV, which can provide a robust bonding between the active material and the substrate to prevent their pulverization [14]. X-ray adsorption spectroscopy (XAS) measurements were also conducted to evaluate the structural evolution from Co(OH)$_2$ to CoOOH. X-ray absorption near-edge structure (XANES) spectra of β-Co(OH)$_2$ and the mixed phase of Co(OH)$_2$/CoOOH are shown in Figure 3C. The intensity of the Co(OH)$_2$/CoOOH mixed phase was much higher than that of β-Co(OH)$_2$, indicating a decreased disorder in the local environment and stronger Co-Co shells [15,16]. The K-edge of the Co(OH)$_2$/CoOOH mixed phase shows a positive shift compared to β-Co(OH)$_2$ in the inset of Figure 3C, which demonstrates an increased Co valence in the mixed phase. In addition, the overall oscillation patterns of β-Co(OH)$_2$ and the mixed phase were quite similar, suggesting similar coordination environments of the Co atom in β-Co(OH)$_2$ and the mixed phase. The extended X-ray absorption fine structure (EXAFS) spectra of β-Co(OH)$_2$ and the mixed phase are displayed in Figure 3D. The similar patterns also suggest similar coordination environments in β-Co(OH)$_2$ and

the mixed phase. Moreover, two peaks were also observed, standing for the Co-O and Co-Co shells. The increased intensities of both peaks in the mixed phase were detected, which indicate a decreased structural disorder in the mixed phase [17,18]. Furthermore, negative shifts of the Co-O and Co-Co shells were also observed in the mixed phase, which suggests the reduced Co-O and Co-Co bond lengths due to the existence of CoOOH. Compared to $Co(OH)_2$, each H atom is shared by the two adjacent Co-O layers in CoOOH. The density functional theory (DFT) calculation simulated the transformation from $Co(OH)_2$ into CoOOH, and its computed energy file is provided in Figure 4. CoOOH (a = b = 3.036 Å; c = 8.862 Å) was obtained by losing one H atom per formula unit from $Co(OH)_2$ (a = b = 3.176 Å; c = 9.358 Å). We found that the energy difference between CoOOH and $Co(OH)_2$ was 0.6 eV, and that the activation energy from CoOOH to $Co(OH)_2$ was higher than for the opposite process, indicating that the mixed phase is more stable than pure $Co(OH)_2$.

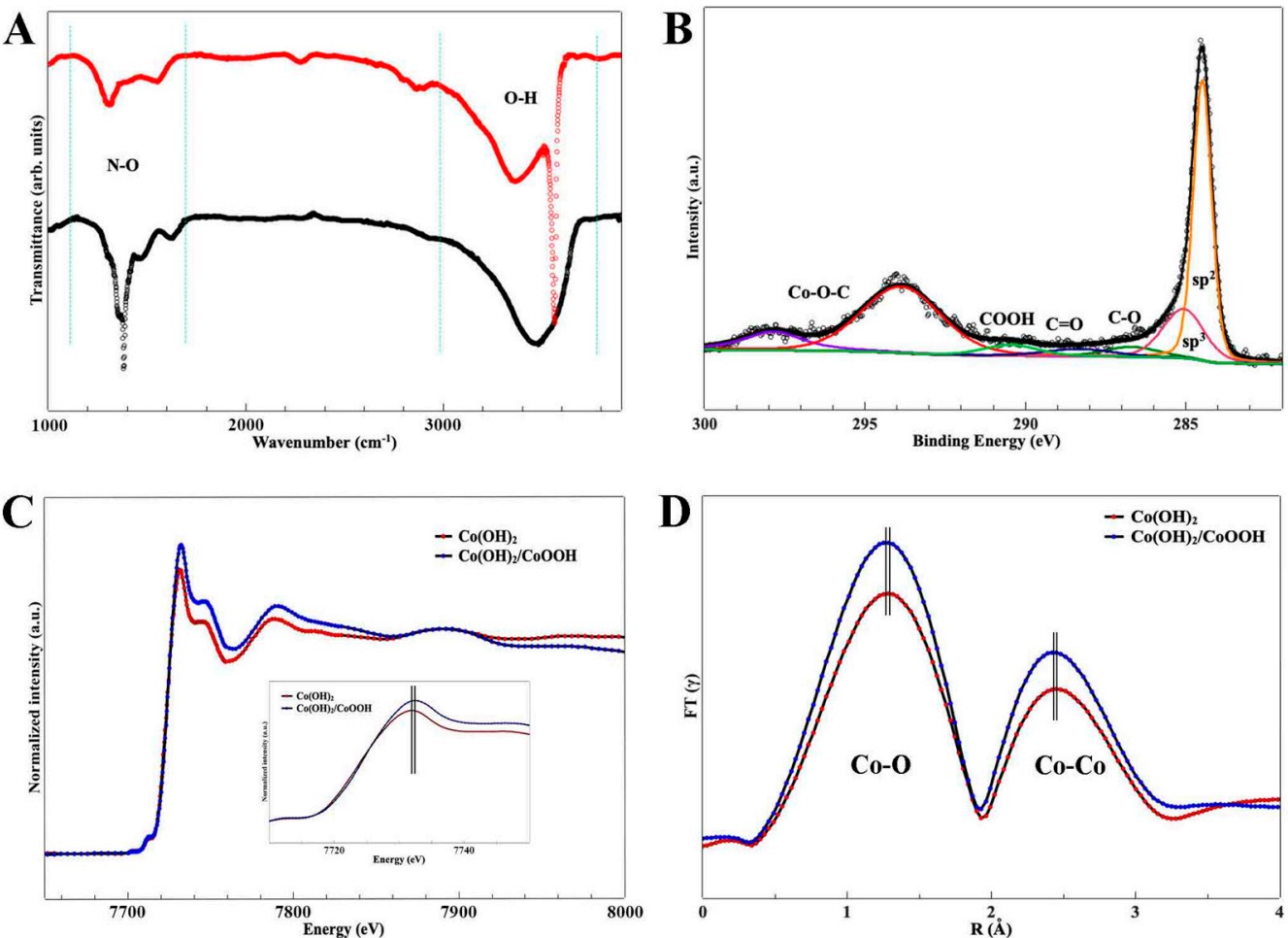

**Figure 3.** (**A**) FTIR analysis of the as-prepared $Co(OH)_2$ and $\beta$-$Co(OH)_2$. Red line: $\alpha$-$Co(OH)_2$; black line: $\beta$-$Co(OH)_2$. (**B**) XPS spectrum of the C 1s of $\alpha$-$Co(OH)_2$/CFP. (**C**) XANES spectra of $\beta$-$Co(OH)_2$ and the mixed phase $Co(OH)_2$ and CoOOH. (**D**) EXAFS spectra of $\beta$-$Co(OH)_2$ and the mixed phase $Co(OH)_2$ and CoOOH. Co-O and Co-Co stand for the nearest Co-O and Co-Co shell to the Co atom.

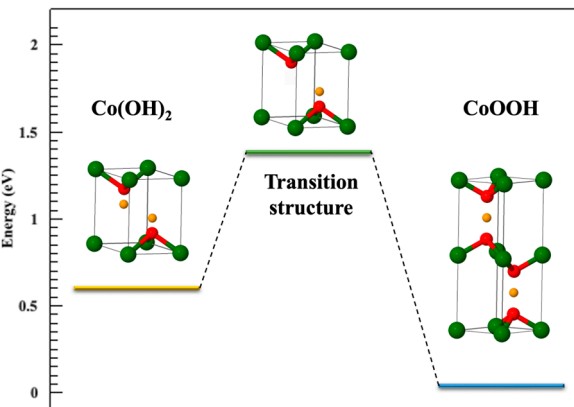

**Figure 4.** Phase transformation energy profile from $Co(OH)_2$ to $CoOOH$. $CoOOH$ is considered as the ground state. Energy values for all the minima and transition states are provided in Table S1. The green, red, and orange balls stand for Co, O, and H atoms, respectively.

The electrochemical performance of the $Co(OH)_2/CoOOH$ mixed phase as the cathode for AZIBs was investigated in a coin cell, in which the $Co(OH)_2/CoOOH$ mixed phase, a Zn foil, and 1 M $ZnSO_4$ solution were used as the cathode, anode, and electrolyte, respectively (Figure 5A). The mixed phase of $Co(OH)_2/CoOOH$ exhibited a maximum capacity of 164 mAh $g^{-1}$ at 0.05 A $g^{-1}$, as well as a high energy density of 275 Wh $kg^{-1}$ (Figure 5B). Moreover, no obvious decay in capacity was observed in the first five cycles, suggesting a good stability of the mixed phase of $Co(OH)_2/CoOOH$. The differential capacity (dQ/dV) curve is depicted in Figure 5C, and two discharge peaks located at 1.6 eV and 1.8 V vs. $Zn^{2+}/Zn$ were observed. The average discharge voltage of 1.7 V vs. $Zn^{2+}/Zn$ was higher than for most of the reported cathode materials, such as $V_2O_5$ (0.6 V vs. $Zn^{2+}/Zn$) and $MnO_2$ (1.4 V vs. $Zn^{2+}/Zn$). Figure 5D shows the Nyquist plot of the mixed phase. The inset is the electrical equivalent circuit, in which $R_{ct}$, $R_s$, W, and CPE stand for the charge-transfer resistance, the equivalent series resistance, the Warburg impedance, and the constant phase element, respectively. According to the fitting results (Table S2), $R_{ct}$ was calculated to be 2.1 $\Omega$, and such a small charge-transfer resistance indicates an excellent electron transfer rate in the mixed phase of $\beta$-$Co(OH)_2$ and $CoOOH$. Figure 5E displays the CV curves of the mixed phase of $Co(OH)_2/CoOOH$ at different scan rates, and the reaction kinetics were evaluated based on the scan rate and the corresponding current. The capacity contribution of the diffusion- and surface-controlled processes can be deconvoluted according to the following equation [19]:

$$i = k_1 v + k_2 v^{\frac{1}{2}}$$

in which $k_1 v$ and $k_2 v^{\frac{1}{2}}$ stand for the surface- and diffusion-controlled processes, respectively. As the scan rate increased from 0.1 mV $s^{-1}$ to 5 mV $s^{-1}$, the contribution ratio of the diffusion-controlled capacity was stabilized at ~70%, indicating that both surface and diffusion synergically regulate the electrode reaction. As a result, the mixed phase of $Co(OH)_2/CoOOH$ exhibited an excellent rate performance. Its capacity can reach 164, 124, 108, 103, 98, and 87 mAh $g^{-1}$ at 0.05, 0.1, 0.2, 0.4, 0.8, and 1.6 A $g^{-1}$, respectively (Figure 5G,H). In contrast, $\beta$-$Co(OH)_2$ only showed a maximum capacity of 48.8 mAh $g^{-1}$ at 0.1 A $g^{-1}$ (Figure S4), along with an inferior rate performance (Figure S5). Compared to recently reported cathode materials for AZIBs, such as PBA (99.7 mAh $g^{-1}$ at 0.05 A $g^{-1}$) [20], FeHCF (76 mAh $g^{-1}$ at 1 A $g^{-1}$) [21], and $Na_3V_2(PO_4)_2F_3$ (65 mAh $g^{-1}$ at 0.08 A $g^{-1}$) [22], the mixed phase of $Co(OH)_2/CoOOH$ exhibited an increased capacity, an improved rate performance, and a higher discharge voltage of 1.7 V (Table 1). The mixed phased also delivered a good stability at the current density of 0.05 A $g^{-1}$. After 200 cycles, a capacity retention of 78% was retained, with high Coulombic efficiencies (Figure 5I). However, the capacity of $\beta$-$Co(OH)_2$ rapidly dropped to 38% after 50 cycles at the current density of 0.1 A $g^{-1}$ (Figure S6). Both $Co(OH)_2$ and $CoOOH$ are amphoteric materials,

which could be dissolved in acidic media. The pH value of the 1 M $ZnSO_4$ electrolyte is about 4~4.5, meaning that the dissolution of the active material is responsible for the capacity loss. On the other hand, the interaction between the $Zn^{2+}$ ions and the Co-O layer was enhanced after the removal of $NO_3^-$ species, which neutralize the positive charge of the Co-O layers. However, the high thermal stability and the shortened Co-Co and Co-O shells, according our XAS analysis and DFT calculation, increased the mechanical strength of the mixed phase to accommodate $Zn^{2+}$ ions and endure the electrostatic repulsion.

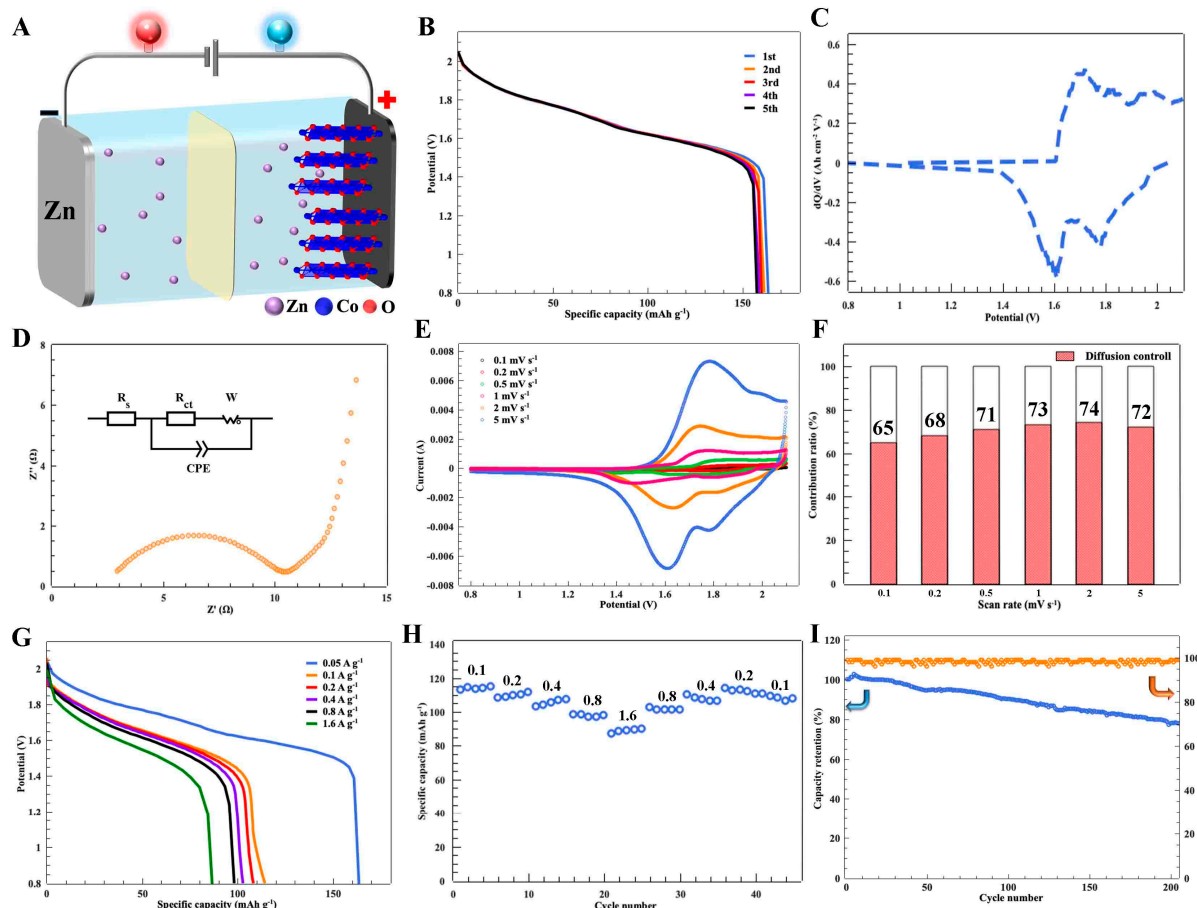

**Figure 5.** (**A**) Schematic illustration of a Zn-Co(OH)$_2$/CoOOH cell in a 1 M ZnSO$_4$ electrolyte. (**B**) Discharge profiles of the first 5 cycles of the mixed phase at 0.05 A g$^{-1}$. (**C**) The differential capacity curves of the mixed phase Co(OH)$_2$/CoOOH. (**D**) The Nyquist plot and the equivalent electrical circuit of the Zn-Co(OH)$_2$/CoOOH cell in a 1 M ZnSO$_4$ electrolyte. (**E**) The CV curves of the mixed phase of Co(OH)$_2$/CoOOH at different scan rates from 0.1 mV s$^{-1}$ to 5 mV s$^{-1}$. (**F**) The contribution ratio of the diffusion-controlled capacity and the surface-controlled capacity. (**G**) Discharge profiles of the mixed phase of Co(OH)$_2$/CoOOH at different current densities from 0.05 A g$^{-1}$ to 1.6 A g$^{-1}$. (**H**) Rate performance of the mixed phase of Co(OH)$_2$/CoOOH. (**I**) The stability (200 cycles at 0.05 A g$^{-1}$) and Coulombic efficiency of the mixed phase of Co(OH)$_2$/CoOOH.

**Table 1.** Electrochemical performance comparison of cathode materials for AZIBs.

| Cathode | Maximum Capacity | Rate Performance | Discharge Voltage | Refs. |
|---|---|---|---|---|
| PBA | 99.7 mAh g$^{-1}$ at 0.05 A g$^{-1}$ | 39.7 mAh g$^{-1}$ at 2 A g$^{-1}$ | 1.4 | [20] |
| FeHCF | 76 mAh g$^{-1}$ at 1 A g$^{-1}$ | 41 mAh g$^{-1}$ at 8 A g$^{-1}$ | 1.5 | [21] |
| Na$_3$V$_2$(PO$_4$)$_2$F$_3$ | 65 mAh g$^{-1}$ at 0.08 A g$^{-1}$ | 33 mAh g$^{-1}$ at 8 A g$^{-1}$ | 1.6 | [22] |
| Co(OH)$_2$/CoOOH | 164 mAh g$^{-1}$ at 0.05 A g$^{-1}$ | 87 mAh g$^{-1}$ at 1.6 A g$^{-1}$ | 1.7 | This work |

## 3. Discussion

Recently, LDHs with abundant H vacancies were obtained via electrochemical methods [23,24], and these LDHs have exhibited promising performance as the cathode material for AZIBs in mild acidic media. Coincidentally, the transition metal oxyhydroxide contains one less H atom than its hydroxide, thus indicating that the excellent performance may originate from oxyhydroxides or the synergic effect of hydroxide and oxyhydroxide. Unfortunately, the effect of oxyhydroxide was overlooked. CoOOH and $Co(OH)_2$ are quite alike in structure, but the oxyhydroxide is more stable, which could stabilize the whole structure against the strong interaction with $Zn^{2+}$ ions. Therefore, according to our work, the transition metal oxyhydroxide is supposed to be a promising cathode material for AZIBs.

## 4. Materials and Methods

Synthesis of $\alpha$-$Co(OH)_2$. The $\alpha$-$Co(OH)_2$ was electrochemically deposited on the carbon fiber paper (CFP) as the substrate. A 1 M $Co(NO_3)_2$ solution (analytical reagent (AR), Xilong Chemical Co., Ltd., Guangzhou, China) was used as the electrolyte. Before the electrodeposition, the CFP was washed with acetone, ethanol, and distilled water for 30 min each, and the CFP was dried in a 60 degree oven overnight. Then, the $\alpha$-$Co(OH)_2$ thin film was electrodeposited on the CFP in a conventional three-electrode electrolytic cell, in which the CFP, a platinum plate, and a saturated calomel electrode (SCE) served as the working, counter, and reference electrode, respectively. The electrodeposition was conducted using a CHI660E potentiostat, and the voltage was maintained at −1 V for 30 min. The formation of $\alpha$-$Co(OH)_2$ can be expressed as follows: $NO_3^- + 7H_2O + 8e^- \rightarrow NH_4^+ + 10\,OH^-$, $Co^{2+} + 2OH^- \rightarrow Co(OH)_2$. Following the electrodeposition, the prepared electrode was washed with distilled water for several times and dried in a 60 degree vacuum oven overnight.

Synthesis of the mixed phase of $Co(OH)_2$/$CoOOH$. The as-prepared $\alpha$-$Co(OH)_2$ was utilized as the precursor to obtain the mixed phase of $Co(OH)_2$ and $CoOOH$ via an electrochemical method in a conventional three-electrode electrolytic cell for a CV treatment, in which the as-prepared $\alpha$-$Co(OH)_2$, a platinum plate, a saturated calomel electrode (SCE), and a 1 M KOH solution served as the working electrode, counter electrode, reference electrode, and the electrolyte, respectively. The CV treatment was conducted using a CHI660E potentiostat. The potential range of the CV treatment was set from −0.1 V to 0.45 V, and the scan rate was 25 mV s$^{-1}$ for 50 cycles. The electrochemical process was expressed as follows: $Co(OH)_2 + OH^- \rightarrow CoOOH + H_2O + e^-$. After the CV treatment, the electrode was washed using the distilled water for several times to wash off the alkaline residues and dried in a 60 degree vacuum oven overnight.

Physical characterization. The morphologies of $\alpha$-$Co(OH)_2$, $\beta$-$Co(OH)_2$, and the mixed phase of $Co(OH)_2$/$CoOOH$ were characterized via scanning electron microscopy (SEM, Hitach SU8000 scanning electron microscope, Hitachi High-Tech, Tokoy, Japan) and transmission electron microscopy (TEM, JEM ARM 1300S for high-resolution TEM images, Thermo Fisher, Shanghai, China). The structural evolution from $\alpha$-$Co(OH)_2$ to the mixed phase of $Co(OH)_2$/$CoOOH$ was probed via X-ray diffraction (XRD, RIGAKU D/MAX2500, Rigaku, Tokoy, Japan). Fourier transform infrared spectra were conducted using a JW-BK132F from the JWGB SCI. & TECH instrument (Beijing, China), and X-ray photoelectron spectroscopy (XPS) spectra were obtained using the ESCALAB-250 instrument with a monochromatic Al K$\alpha$ radiation source and a hemisphere detector with an energy resolution of 0.1 eV. The X-ray absorption (XAS) spectra were obtained on the beamline 1W1B at the Beijing Synchrotron Facility (BSRF), with an electron of 2.2 GeV and a beam current of 250 mA.

Electrochemical evaluation. The electrochemical performances of $\beta$-$Co(OH)_2$ and the mixed phase were carried out in a coin cell, in which Zn foil and 1 M $ZnSO_4$ served as the anode and the electrolyte, respectively. The cathode and anode were separated via a glass microfiber filter (CAT No. 1823-125). The galvanostatic profiles, dQ/dV plot, and cycling

performance were conducted using a Neware battery testing system. The CV curves and Nyquist plot were obtained using a CHI660E potentiostat.

DFT simulations. The Perdew–Burke–Ernzerhof (PBE) functional and supercell approach were utilized as implemented in the Vienna ab initio simulation package (VASP) to perform the spin-polarized DFT calculations. Co (3s, 3p, 3d, and 4s) and O (2s and 2p) electrons were treated as valence states, with a cut-off energy of 520 eV in plane waves, and PBE-based projector-augmented wave potentials were used to replace the other electrons. All energies were computed using the DFT + U method. $Co_2O_4H_2$ (two formula units) was used as a unit cell, and the equilibrium lattice constants were simulated with cell shape, a lattice vector, and atomic position using a residual force of 0.02 eV $Å^{-1}$. These conditions ensured a convergence in the equilibrium distance. As for the transition state, the considered phase transformation was located using the elastic band algorithm.

**5. Conclusions**

To sum up, we synthesized a mixed phase of $Co(OH)_2/CoOOH$ via a two-step electrochemical preparation. The inserted $NO_3^-$ species was removed in the alkaline media, resulting in the transformation from α- into β-$Co(OH)_2$, and CoOOH was obtained based on the energy storage mechanism of $Co(OH)_2$ in supercapacitors. As the cathode material for AZIBs, the electrochemical performance of the mixed phase of $Co(OH)_2/CoOOH$ was much improved than that of β-$Co(OH)_2$. The mixed phase delivered a maximum capacity of 164 mAh $g^{-1}$ at 0.05 A $g^{-1}$, a high average discharge plateaus of 1.7 V, as well as a high energy density of 275 Wh $kg^{-1}$. Benefiting from the low charge-transfer resistance, $Co(OH)_2/CoOOH$ showed a good rate performance, maintaining a capacity of 87 mAh $g^{-1}$ at 1.6 A $g^{-1}$. DFT calculations and various spectroscopy analyses suggested the high thermal stability of the mixed phase of $Co(OH)_2/CoOOH$ than pure $Co(OH)_2$ because of less local structural disorder. The reduced Co-Co and Co-O shells increased the mechanical strength of the mixed phase to accommodate $Zn^{2+}$ ions and endure the electrostatic repulsion, resulting in an enhanced cycling stability. This work provides a new perspective to fabricating LDH materials for high-performance AZIBs.

**Supplementary Materials:** The following supporting information can be downloaded at: https://www.mdpi.com/article/10.3390/inorganics11100400/s1, Figure S1: the XRD pattern of CFP; Figure S2: the SEM image after the CV treatment; Figure S3: the TEM image after the CV treatment; Figure S4: discharge profiles of β-$Co(OH)_2$ at 0.1 A $g^{-1}$; Figure S5: the rate performance comparison of β-$Co(OH)_2$ and the mixed phase; Figure S6: the stability test of β-$Co(OH)_2$ at 0.1 A $g^{-1}$; Table S1: total energy values (in eV) for the minima and the transition state shown in Figure 4; Table S2: the fitting results of the EIS spectra of the mixed phase of $Co(OH)_2$ and CoOOH.

**Author Contributions:** Conceptualization, T.D.; Funding acquisition, T.D.; Investigation, F.L. and Y.Z.; Resources, T.D.; Supervision, T.D.; Validation, H.U.; Visualization, F.L.; Writing—original draft, F.L. and Y.Z.; Writing—review and editing, H.U. and T.D. All authors have read and agreed to the published version of the manuscript.

**Funding:** This research was funded by the International Collaboration Program of Jilin Provincial Department of Science and Technology (No. 20230402051GH), the Open Project Program of Key Laboratory of Preparation and Application of Environmental Friendly Materials (Jilin Normal University) of the Ministry of China (No. 2021006), the Fundamental Research Funds for the Central Universities JLU, and "Double-First Class" Discipline for Materials Science and Engineering.

**Data Availability Statement:** The data presented in this study are available on request from the corresponding author. The data are not publicly available due to the data protection.

**Acknowledgments:** We acknowledge Beijing Synchrotron Radiation Facility (BSRF) for their XAS measurements on the beamline 1W1B.

**Conflicts of Interest:** The authors declare no conflict of interest. The funders had no role in the design of the study.

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
