# Peer review of "An Electrochemically Prepared Mixed Phase of Cobalt Hydroxide/Oxyhydroxide as a Cathode for Aqueous Zinc Ion Batteries"

_inorganics, doi:10.3390/inorganics11100400_

Round 1

Reviewer 1 Report

The authors have submitted a manuscript entitled “Electrochemically Prepared Mixed Phase of Cobalt Hydroxide/Oxyhydroxide as Cathode for Aqueous Zinc Ion Batteries”. The manuscript is interesting. However I would like to ask few questions and provide a few suggestions.

Comments

1)      It is important to consistently introduce acronyms the first time they are used in the manuscript. For instance, in the abstract, the acronym "AZIB" should be clarified upon its first appearance.

2)       There is an issue with the sentence, "The safety issue and high cost of lithium-ion batteries have driven us to search for alternative energy storage devices (ESDs) with higher cost-performance." The phrase "higher cost-performance" appears to be a typo or a miswording. Is that Higher performance or something else? Please clarify the intended meaning of this sentence.

3)      In the XRD pattern of 1B, it is recommended to use the notation (001), (012), and (018) for the crystallographic planes instead of just "C(001)," "C(012)," and "C(018)." If you choose to retain the current notation, ensure that you provide a clear explanation in the text.

4)      The statement, "Co(OH)2 displayed a petal- like morphology with an average length of 1 µm and thickness of 500 nm (Figure 2A)," appears to conflict with the information provided in the accompanying image and scale bar. It seems the thickness is less than that of the authors mentioned. To improve accuracy, it is advisable to employ image processing software like ImageJ to precisely measure the thickness. Additionally, marking the thickness directly on the SEM image would enhance clarity.

5)      In Figure 2D's TEM image, there are four "V" shapes that require explanation in the text. Please provide this explanation.

6)      It is not clear whether SEM or TEM images for Co(OH)2/CoOOH have been provided. If they have been included, this should be explicitly stated. If not, the authors might consider including them.

7)      The description of the FTIR analysis is lacking clarity. It is unclear whether the authors have conducted FTIR measurements for Co(OH)2/CoOOH samples. This aspect needs clarification.

8)      As the authors focus on the sample Co(OH)2/CoOOH only for final applications, the authors should provide the given characterization results for Co(OH)2/CoOOH along with the other samples.

Author Response

Reviewer #1: The authors have submitted a manuscript entitled “Electrochemically Prepared Mixed Phase of Cobalt Hydroxide/Oxyhydroxide as Cathode for Aqueous Zinc Ion Batteries”. The manuscript is interesting. However, I would like to ask few questions and provide a few suggestions.

Answer: Many thanks for your evaluation of our work. We made revisions according to your comment in this version. In the following we answer to the comments.

1, It is important to consistently introduce acronyms the first time they are used in the manuscript. For instance, in the abstract, the acronym "AZIB" should be clarified upon its first appearance.

Answer: Many thanks for your comment. In this version, we carefully check through the manuscript to eliminate this cursoriness.

2, There is an issue with the sentence, "The safety issue and high cost of lithium-ion batteries have driven us to search for alternative energy storage devices (ESDs) with higher cost-performance." The phrase "higher cost-performance" appears to be a typo or a miswording. Is that Higher performance or something else? Please clarify the intended meaning of this sentence.

Answer: Many thanks for your comment. The wide application of lithium-ion batteries originates from their proven techniques and high performances. However, the content of Li source is limited when compared with other metallic elements. What we intended to express was that searching alternative batteries with comparable performance but lower cost is quite important for the sustainable development. We realized the ambiguity of this sentence, so in this version, we have rephrased this sentence: “The safety issue and high cost of lithium-ion batteries have driven us to search for alternative energy storage devices (ESDs) with comparable performance, but higher safety and lower cost.

3, In the XRD pattern of 1B, it is recommended to use the notation (001), (012), and (018) for the crystallographic planes instead of just "C(001)," "C(012)," and "C(018)." If you choose to retain the current notation, ensure that you provide a clear explanation in the text.

Answer: Thanks a lot for your comment, and we have made according explanation in the text and Figure 1B in Page 2.

4, The statement, "Co(OH)2 displayed a petal-like morphology with an average length of 1 µm and thickness of 500 nm (Figure 2A)," appears to conflict with the information provided in the accompanying image and scale bar. It seems the thickness is less than that of the authors mentioned. To improve accuracy, it is advisable to employ image processing software like ImageJ to precisely measure the thickness. Additionally, marking the thickness directly on the SEM image would enhance clarity.

Answer: Thanks to your comment, we have realized this error and we have revised Figure 2 and the according description.

5, In Figure 2D's TEM image, there are four "V" shapes that require explanation in the text. Please provide this explanation.

Answer: Many thanks for your comment. The space group of hexagonal b-Co(OH)2 is R3m, which implies that the crystal planes of (100) and (010) are placed with an acute angle of 60°. On the other hand, the acute 60° angle can also manifest the formation of  b-Co(OH)2.

6, It is not clear whether SEM or TEM images for Co(OH)2/CoOOH have been provided. If they have been included, this should be explicitly stated. If not, the authors might consider including them.

Answer: Many thanks for your comment. In the last version, we have provided SEM and TEM images of the mixed phase of Co(OH)2 and CoOOH in Figure S2 and Figure S3. Compared to the morphology of b-Co(OH)2, the mixed phase showed a similar morphology due to structure similarity of Co(OH)2 and CoOOH, so that we placed the SEM and TEM images in the SI. According to your comment, we have revised the statement of the morphology of the mixed phase in Page 3.

7, The description of the FTIR analysis is lacking clarity. It is unclear whether the authors have conducted FTIR measurements for Co(OH)2/CoOOH samples. This aspect needs clarification.

Answer: Many thanks for your comment. The reason for the FTIR analysis is multifold.

First of all, a-Co(OH)2 is a phase with ion/molecule inserted between adjacent Co-O layers, so that FTIR can examine the inserted species due to specific vibration modes. Secondly, a-Co(OH)2 is quite unstable in the alkaline media, which can rapidly transform to b-Co(OH)2 proven by our previous work and this work. The reduced intensity of N-O stretching mode implies the removal of the inserted NO- 3 and the compact Co-O layer arrangement of b-Co(OH)2. Finally, the FTIR analysis also manifest that the actual reactant is b-Co(OH)2 rather than a-Co(OH)2. In this version, we have added clarification about the FTIR analysis in Page 4.

8, As the authors focus on the sample Co(OH)2/CoOOH only for final applications, the authors should provide the given characterization results for Co(OH)2/CoOOH along with the other samples.

Answer: Many thanks for your comment. In this version, we have compared the mixed phase with pure b-Co(OH)2 and recently reported cathode materials for AZIBs, and according revisions are made.

Reviewer 2 Report

1.     For EIS data (Fig. 5C), the fitting results should be displayed together with the experimental data. In addition, all the circuit parameters should be obtained, discussed, and compared with the literature.

2.     The cyclic voltammetry measurements should be performed at different scan rates.

  1. All findings (capacitance, energy density, etc.) should be compared with other related materials. A detailed table of comparison should be provided.
  2. The correlation between structural/morphological findings and electrochemical performance should be discussed and compared with other related composites.
  3. The stability test should be performed for a longer time, 10000 cycles.

Moderate editing of English language required

Author Response

Review #2:

1, For EIS data (Fig. 5C), the fitting results should be displayed together with the experimental data. In addition, all the circuit parameters should be obtained, discussed, and compared with the literature.

Answer: Many thanks for your comment. The fitting results are placed in Figure 5C and all the circuit parameters are listed in Table S2. According to the fitting results, the small charge-transfer resistance of 2.1 W indicates an excellent electron transfer rate of the mixed phase of b-Co(OH)2 and CoOOH.

2, The cyclic voltammetry measurements should be performed at different scan rates.

Answer: Many thanks for your comment. CV measurements at different scan rates were conducted and the results are shown in the new Figure 5E. And the contribution ratio of the diffusion/surface-controlled processes is depicted in Figure 5F. The diffusion-controlled capacity contributed about 65% at the scan rate of 0.1 mV s-1. As the scan rate increased, the diffusion-controlled contribution was stabilized at 70%, indicating that the charge storage process was synergically controlled by the diffusion and the surface.

3, All findings (capacitance, energy density, etc.) should be compared with other related materials. A detailed table of comparison should be provided.

Answer: Many thanks for your comment. In this version, we have compared Co(OH)2/CoOOH with recently reported cathode materials, such as PBA, FeHCF and Na3V2(PO4)2F3, the performance of which are compared and listed in Table 1 and according revisions are made.

4, The correlation between structural/morphological findings and electrochemical performance should be discussed and compared with other related composites.

Answer: Many thanks for your comment. In this version, we have compared the mixed phase with pure b-Co(OH)2 in Figure S4, S5 and S6, and according revisions are made.

5, The stability test should be performed for a longer time, 10000 cycles.

Answer: Many thanks for your comment. A long lifespan is quite necessary for next generation energy storage devices. As we stated before, Co(OH)2 and CoOOH are amphoteric, which are quite sensitive to the working condition. The electrolyte of this work is 1 M ZnSO4 and the pH is about 4 ~ 4.5, which is responsible for the capacity loss. Till now, the best energy retention we could obtain is 78% after 200 cycles. After that the capacity would drop rapidly due to the dissolution of the active material.

Round 2

Reviewer 2 Report

The authors have addressed most of my comments in the revised manuscript. The manuscript could be published in Inorganics.

 Minor editing of English language required